# A Dynamic Load Simulation Algorithm Based on an Inertia Simulation Predictive Model

Kuizhi Lin [1,2], Wei Wang [1,2,*], Zhiqiang Wang [1,3], Xinmin Li [1,3] and Huang Zhang [1,4]

1   National Joint Local Engineering Research Center of Electrical Machine System Intelligent Design and Manufacturing, Tianjin 300387, China; linkuizhi@tiangong.edu.cn (K.L.); wangzhiqiang@tiangong.edu.cn (Z.W.); lixinmin@tju.edu.cn (X.L.); zhanghuang0415@163.com (H.Z.)
2   School of Electronics and Information Engineering, Tiangong University, Tianjin 300387, China
3   School of Electrical Engineering, Tiangong University, Tianjin 300387, China
4   School of Mechanical Engineering, Tiangong University, Tianjin 300387, China
*   Correspondence: wangweibit@163.com

**Abstract:** In this study, an electric dynamic load simulation system (EDLSS) algorithm was proposed based on an inertia simulation predictive model to mitigate strong-coupling torque disturbance and load torque fluctuation caused by the change in the motion state of a bearing system. First, the inertia simulation model was proposed by combining the dynamic equations of both the EDLSS and the target system. The aforementioned inertia simulation model converted the conventional realisation method of the inertia simulation into the tracking of the motion characteristics for the target system under the same working conditions. Next, based on the aforementioned inertia simulation model while considering the strong-coupling torque effect and motor braking state disturbance as two influential factors, an inertia simulation predictive model and a load simulation algorithm were proposed. The predicted speed calculated by the predictive model was consistent with the dynamic characteristics of the target system under the same working conditions and input into the control loop. Based on the analysis of the braking state and power model of the permanent magnet synchronous motor, an energy feedback control method was proposed to improve EDLSS stability caused by the braking state of the loading motor. Finally, the experimental data revealed that the maximum speed fluctuation range of the loading motor was approximately 7.5, which was 84% lower than the range before the application of the aforementioned algorithm, which was about 46.8. Furthermore, the maximum range of the torque ripple was close to 1.5, which was 75% lower than before, which was roughly 6. All experimental data were consistent with simulation data.

**Keywords:** electric dynamic load simulation system (EDLSS); inertia simulation; load simulation; regenerative braking; permanent magnet synchronous motor (PMSM)





## 1. Introduction

Permanent magnet synchronous motors (PMSMs) are widely used in electric dynamic load simulation systems (EDLSSs) owing to their high power factor, high power density, and high operating efficiency. EDLSSs are used to perform semi-physical simulations to test the performance indicators of servo systems. When EDLSSs are used for the servo system load test, the loading motor is dragged by the bearing system to follow its synchronous motion and simulate the load and inertia torques. During this period, the loading motor is affected by the motion state of the bearing system to produce load torque fluctuation [1–3]. Therefore, suppressing the strong-coupling torque disturbance of the servo system to the EDLSS is crucial.

In current research, interference torque suppression is typically realised using the following aspects. On the loading motor control side, the loading motor controller can be tuned, the control parameters can be optimised, and the control error can be decreased. Furthermore, the disturbance torque generated by the loading motor side can be eliminated.

Some model predictive control methods have been proposed in [4–7]. In [4], an improved three-vector model predictive control strategy was proposed. In this model, the principle of deadbeat synchronisation between torque and flux linkage was adopted to reduce six candidate vectors in conventional torque prediction to two, and a value function was designed to select the optimal voltage vector to obtain small torque ripple and current harmonics. In [5], a model predictive current control strategy was proposed. In this model, the stator current predictive model was derived, and the first and second optimal voltage vectors and their respective action times were calculated using the value function. These values were then output directly to the inverter so that the torque ripple was reduced by 9.40% and 4.80%, respectively. In [6], a modified model predictive control method with the virtual model was proposed to reduce the dynamic tracking error of the position loop and the dynamic tracking error and stability time by generating a virtual reference. Adaptive control methods were adopted in [8–12]. In [8], a feedforward control scheme based on an adaptive observer was proposed for tracking problems with unknown mechanical parameters and unknown load torque. The adaptive law of inertia was deduced and Popov's theory was introduced into the stability analysis of the adaptive observer to improve the dynamic response performance of the controller. In [9], a control method for the PMSM based on adaptive dynamic programming was proposed to establish two neural networks to control torque and realise low torque ripple under an unknown load. In [10], an adaptive optimal control method was proposed for the dynamic disturbance of velocity and current. The conventional velocity and current tracking control problem was transformed into an optimal control problem, and the optimal solution was obtained by the Hamilton–Jacobi–Isaacs equation so that the $d$- and $q$-axes current disturbance tends to be 0 under the effect of external disturbance torque.

From the perspective of the position error between the servo and the loading systems, a compensation signal was introduced to mitigate the disturbance of the position signal of the servo system caused by the disturbance torque for realising a synchronous movement of the loading motor and the bearing system [13–18]. In [13], an adaptive linear active disturbance rejection control controller was proposed. The suppression performance of the system for disturbance torque and position noise was improved by changing the series structure of the position and velocity in the conventional adaptive linear active disturbance rejection control loop into a parallel structure and combining the linear full-order extended state observer with the control loop. In [14], a fractional-order extended state observer was combined with the active disturbance rejection controller and subsequently applied to the position loop of the servo system. The disturbance was eliminated before the output of the system was affected by internal and external disturbances. Compared with integer-order proportional-integral-differential and linear-active disturbance rejection controllers, the method has high bandwidth, fast response speed, and excellent anti-interference performance.

When the EDLSS loads the servo system, the loading motor functions in the power generation state. By reversing current $i_q$, magnetic resistance is applied to the loading motor under the action of reverse drag, and the process of simulating the load torque and inertia torque can be approximated as braking of the loading motor. At this stage, braking is classified into two categories, namely energy feedback and energy consumption braking [19–22]. The experimental results revealed that when the loading motor enters the energy consumption braking state, the additional energy consumption causes the driver that is coupled directly with the loading motor to produce undervoltage because of the braking state. This phenomenon affects EDLSS stability.

The aforementioned schemes introduce feedforward compensation to eliminate disturbance in the control loop according to the sampled position signal, realise synchronous motion, and suppress interference. However, when the position signal passes through the high-order differential term of the observer, the error is amplified to form interference. We strove to design a dynamic load simulation algorithm that can suppress the strong-coupling torque and load torque fluctuation during the loading process of EDLSSs.

In this study, a load simulation algorithm was proposed based on the inertia simulation predictive model. First, a novel inertia simulation model was proposed. In this model, the dynamic equations of the EDLSS and the target system were combined so that the conventional realisation method for inertia simulation was converted into the tracking of the motion characteristics of the target system under the same driving torque. Next, based on the aforementioned inertia simulation model, a novel inertia simulation predictive model and a load simulation algorithm were proposed. By considering the strong-coupling torque disturbance and the motor braking state disturbance as the two influencing factors in the model and using the non-catastrophic property of the inertia system, the predicted speed that satisfies the dynamic characteristics of the target system under the same working conditions was obtained. The braking state and power model of PMSM were analysed, and an energy feedback control method was proposed to improve EDLSS stability caused by the braking state of the loading motor. Finally, the simulation analysis and experimental verification were performed in MATLAB/Simulink on a self-developed experimental platform.

## 2. Nonsalient-Pole PMSM Model

A *d-q* synchronous rotating coordinate system was established on the rotor, and Clarke and Park transforms of three-phase current vector $i_a i_b i_c$ were performed to obtain vector projection $i_d i_q$ in a *d-q* synchronous rotating coordinate system.

Considering the magnetic coupling effect and ignoring the field-weakening control, the flux model, voltage model, and torque model of PMSM were established in a *d-q* synchronous rotating coordinate system.

The flux linkage equation of a PMSM in a *d-q* synchronous rotating coordinate system can be expressed as follows:

$$\begin{bmatrix} \psi_d \\ \psi_q \end{bmatrix} = \begin{bmatrix} L_d & 0 \\ 0 & L_q \end{bmatrix} \begin{bmatrix} i_d \\ i_q \end{bmatrix} + \begin{bmatrix} \psi_f \\ 0 \end{bmatrix}, \tag{1}$$

where $\psi_d$ is the stator flux linkage component in the *d*-axis of a synchronous rotating coordinate system; $\psi_q$ is the stator flux component in the *q*-axis of a synchronous rotating coordinate system; $\psi_f$ is the flux linkage generated by the permanent magnet; $L_d$ is the inductance in the *d*-axis of a PMSM; $L_q$ is the inductance in the *q*-axis of a PMSM; $i_d$ and $i_q$ are the vector projections of three-phase current vector $i_a i_b i_c$ in a *d-q* synchronous rotating coordinate system obtained by Clarke and Park transforms.

The vector relationship of each magnetic chain component in the flux linkage model of a PMSM is displayed in Figure 1, where $\psi_s$ is the vector resultant of each magnetic chain component in a *d-q* synchronous rotating coordinate system.

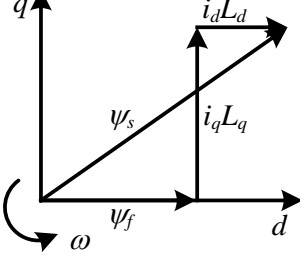

**Figure 1.** The flux linkage model of a PMSM.

The voltage equation of the PMSM in a *d-q* synchronous rotating coordinate system can be expressed as follows:

$$\begin{bmatrix} u_d \\ u_q \end{bmatrix} = R \begin{bmatrix} i_d \\ i_q \end{bmatrix} + P \begin{bmatrix} \psi_d \\ \psi_q \end{bmatrix} + \omega \begin{bmatrix} -\psi_q \\ \psi_d \end{bmatrix}, \tag{2}$$

where $u_d$ and $u_q$ are the vector projections of three-phase voltage vector $u_a u_b u_c$ in a *d-q* synchronous rotating coordinate system after Clarke and Park transforms; *R* is the stator resistance; *P* is a differential operator; and $\omega$ is the rotor angular velocity.

The simultaneous Equations (1) and (2) can be solved as follows:

$$u_d = Ri_d + L_d \frac{di_d}{dt} - \omega L_q i_q, \tag{3}$$

$$u_q = Ri_q + L_q \frac{di_q}{dt} + \omega \left( L_d i_d + \psi_f \right). \tag{4}$$

The voltage–current vector relationship is displayed in Figure 2, where $u_s$ is the vector resultant of voltage components in a *d-q* synchronous rotating coordinate system, $i_s$ is the vector resultant of current components in a *d-q* synchronous rotating coordinate system.

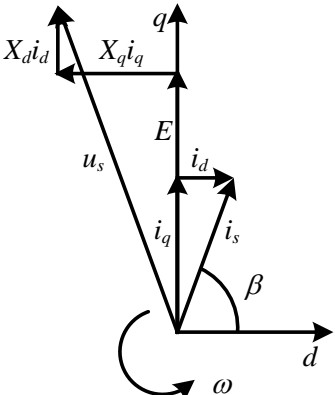

**Figure 2.** The voltage–current model of a PMSM.

The torque equation of the three-phase PMSM can be expressed as follows:

$$\begin{aligned} T_e &= p_n \left( \psi_d i_q - \psi_q i_d \right) \\ &= p_n \left[ \psi_f i_q + \left( L_d - L_q \right) i_d i_q \right], \end{aligned} \tag{5}$$

where $p_n$ is the pole pairs.

Equation (5) reveals that the electromagnetic torque consists of the following two parts:

$$T_{e1} = p_n \psi_f i_q \tag{6}$$

$$T_{e2} = p_n \left( L_d - L_q \right) i_d i_q. \tag{7}$$

Similarly, Equation (5) can be expressed as follows:

$$T_e = T_{e1} + T_{e2}, \tag{8}$$

where Equation (6) is the electromagnetic torque generated by the interaction between the electromagnetic field and the permanent magnet, namely the excitation torque; Equation (7) is the torque caused by the uneven magnetic resistance of the magnetic circuit caused by the saliency effect, namely the magnetic resistance torque.

For a nonsalient-pole PMSM, no salient pole exists in its rotor structure, that is, $L_d = L_q$. Therefore, the magnetic resistance torque does not exist in Equation (7), that is, $T_{e2} = 0$. In this case, only the single effect of the excitation torque should be considered. Thus, the torque equation of a nonsalient-pole PMSM can be obtained as follows:

$$T_e = T_{e1} = p_n \psi_f i_q = p_n \psi_f i_s \sin \beta, \tag{9}$$

where $\beta$ is the phase angle between $i_s$ and the *d*-axis in a *d-q* synchronous rotating coordinate system.

For a PMSM, $\psi_f$ generated by the permanent magnet is fixed. Therefore, according to Equation (9), the excitation torque is only related to $i_q$, the vector projection of the three-phase current vector in the *q*-axis, and when the phase angle $\beta$ is $\pi/2$, the excitation torque has a maximum value. In this case, $i_s$ coincides with the *q*-axis, that is, $i_d = i_s cos\beta = 0$.

### 3. Inertia Simulation Model

In the EDLSS, inertia simulation is typically realised using the load torque. The load and inertia torque simulated by the loading motor are tracked by controlling the current $i_q$, which causes the simulation system to be consistent with the dynamic characteristics of the target system.

Compared with the load torque, the friction torque of the mechanical structure is too small to be ignored. Assuming that the dynamic equation of the target system in theory is as follows:

$$T_D - T_{basic} = J_s \frac{d\omega}{dt}, \tag{10}$$

where $T_D$ is the driving torque; $T_{basic}$ is the basic load torque; $J_s$ is the moment of inertia of the target system; and $d\omega/dt$ is the angular acceleration.

For the same reason, the dynamic equation of the EDLSS can be assumed as follows:

$$T_D - T_m = J_m \frac{d\omega}{dt}, \tag{11}$$

where $T_m$ is the load torque of motor output; and $J_m$ is the moment of inertia of the simulation system.

At this stage, if the aforementioned two systems are under the same driving torque, the dynamic equation of the inertia simulation system can be expressed as follows:

$$T_m - T_{basic} = (J_s - J_m) \frac{d\omega}{dt}. \tag{12}$$

If the simulation system and the target system have the same $\omega$ and $d\omega/dt$ under the same driving torque, the simulation system satisfies the same dynamic characteristics as the target system by simulating the load torque and inertia torque of the target system.

According to the aforementioned analysis, by establishing the inertia simulation model of the simulation system, the realisation method of inertia simulation can be transformed from 'tracking the load torque and inertia torque required for the simulation of the loading motor' to 'tracking the motion characteristics of the target system under the same driving torque', such as the angular velocity and angular acceleration.

The inertia simulation structure is displayed in Figure 3.

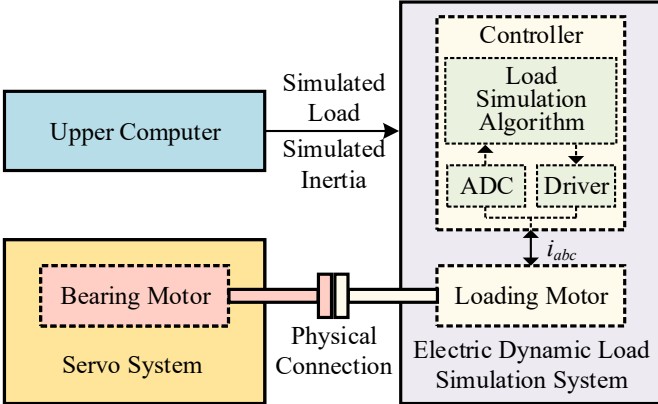

**Figure 3.** The inertia simulation structure.

In the figure, the bearing motor and the loading motor are connected by the mechanical structure, and the bearing motor drives the synchronous movement of the loading motor. The bearing motor, loading motor, and controller constitute a fast-response closed-loop torque control system. The dynamic compensation torque is generated based on the acceleration variation of the loading motor, which is used to compensate for the torque error caused by the difference between the inertia of the simulation system and the inertia of the target system under the same driving torque. The dynamic compensation torque is converted into the control signal of the inverter so that the error between the output torque and the target torque tends to be 0.

## 4. Regenerative Braking

When the EDLSS loads the servo system, the loading motor is in the power generation state. The process of simulating the load and inertia torques can be approximated as the braking of the loading motor through reversing current $i_q$ and applying the magnetic resistance to the loading motor under the action of back drag. At this stage, the braking is classified into two categories, namely energy feedback braking and energy consumption braking. When the back electromotive force is sufficient to provide the braking current, the loading motor is in the energy feedback braking state; when the back electromotive force is insufficient to provide the required braking current, the loading motor requires an additional braking current, which is in the braking state of energy consumption.

Experimental results revealed that when the loading motor is in the energy consumption braking state, the additional energy consumption renders the driver that is coupled directly with the loading motor to undervoltage because of the effect of the braking state. Thus, the stability of the EDLSS is affected.

Based on a synchronous rotating coordinate system, the input power of the PMSM can be expressed as follows:

$$P_{in} = u_d i_d + u_q i_q. \tag{13}$$

Equations (3), (4) and (13) can be solved simultaneously as follows:

$$P_{in} = \left(R i_d - \omega L_q i_q\right) i_d + \left(R i_q + \omega L_d i_d + \omega \psi_f\right) i_q. \tag{14}$$

If the field-weakening control and the iron loss current of the $d$-axis are ignored, the maximum torque control is adopted, that is, $i_d = 0$.

Assuming that $\omega$, the electric speed of the rotor, is a constant, then the input power $P_{in}(i_q)$ is a quadratic function of the current $i_q$, from which the power model of PMSM in the steady state is obtained as follows:

$$P_{in} = \left(\omega \psi_f + R i_q\right) i_q \tag{15}$$

The input power function diagram of PMSM at various $\omega$ values is displayed in Figure 4.

In the figure, the positive direction of $i_q$ is its direction in the motor drive state. Let $P_{in} = 0$ to obtain two zeros of $P_{in}(i_q)$ at $i_q = -\omega \psi_f/R$ and $i_q = 0$. When $i_q < -\omega \psi_f/R$, $P_{in} > 0$. At this stage, the braking requires a positive input power, which indicates that the motor functions in the state of energy consumption braking. When $-\omega \psi_f/R < i_q < 0$, $P_{in} < 0$. At this stage, the input power is negative, which indicates that the motor operates in the energy feedback braking state. When $i_q = -\omega \psi_f/2R$, a minimum value exists in $P_{in}(i_q)$, indicating that the energy feedback of motor braking has a maximum value.

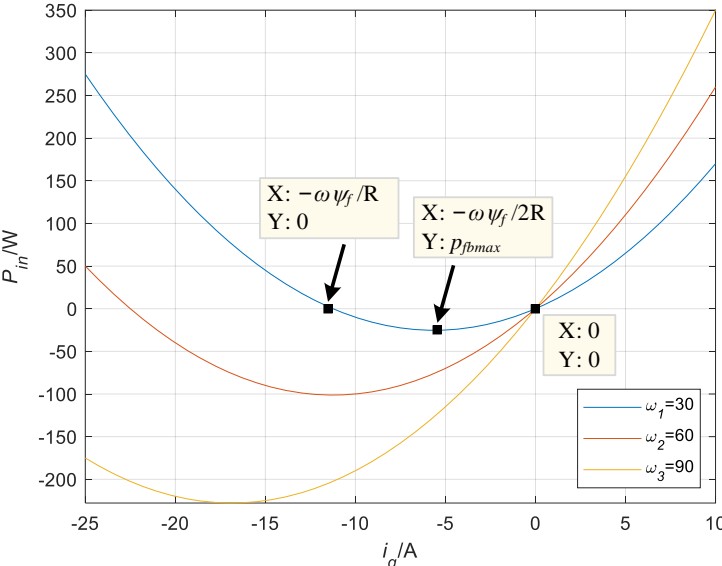

**Figure 4.** The input power function diagram of PMSM at various $\omega$ values ($\psi_f$ = 0.3 Wb, $R$ = 0.8 $\Omega$).

Based on the aforementioned analysis of the braking state and the power model, an energy feedback control method was proposed to improve EDLSS stability caused by the braking state of the loading motor, as displayed in Figure 5. In the figure, $dcK_p$ and $dcK_i$ are the proportional and the integral coefficients of the energy feedback proportional integral (PI) controller; $\omega_r$ is the angular velocity of the motor rotor; $\tau_{dc}$ is the time constant of the filter; $K_\omega$ is the regenerative braking factor, which can be expressed as follows:

$$K_\omega = \frac{C_1 \omega_r \psi_f}{R},\qquad(16)$$

where $C_1$ is a fixed coefficient.

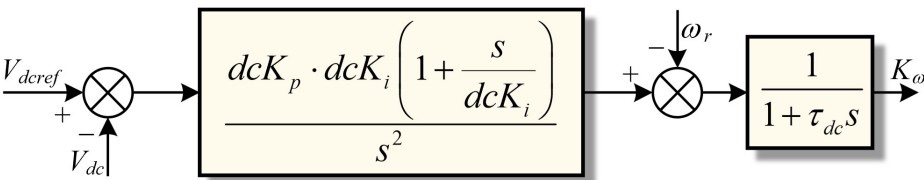

**Figure 5.** The energy feedback control method.

## 5. Inertia Simulation Predictive Model

In addition to the effect of the braking state of the loading motor, when the EDLSS is used for an on-load test of a servo system, the disturbance torque introduced by the loading motor in response to the change in the motion state of the bearing system renders achieving synchronous motion with the bearing system difficult, which results in torque fluctuation. Therefore, the smooth load simulation cannot be output.

Such load fluctuations caused by position disturbance can be suppressed by compensating the position signal of the servo system to realise the synchronous motion of the loading motor and the bearing system. However, external sensors are required and other interference can be easily introduced because of its complex calculation. Thus, the load simulation system loses robustness.

Based on the inertia simulation model of the aforementioned simulation system, an inertia simulation predictive model and a load simulation algorithm were proposed. The speed of the loading motor that conforms to the dynamic characteristics at a continuous time is predicted by analysing the dynamic characteristics of the loading motor under the aforementioned two main disturbances when simulating the inertia torque and the load

torque and using the non-catastrophic property of the inertia system to suppress the load torque fluctuation caused by position disturbance.

The dynamic equation of the aforementioned inertia simulation model can be rewritten considering strong-coupling torque effect and motor braking state disturbance as two influencing factors of the model to obtain the following expression:

$$T_m - K^2{}_{accbasic}(T_{basic} + K_\omega) = (J_s - J_m)\frac{\omega_{k+1} - \omega_k}{T_s}, \tag{17}$$

where $\omega_{k+1}$ is the $k + 1$ moment predicted rotor angular velocity; $\omega_k$ is the $k$ moment rotor angular velocity; $T_s$ is the system control cycle; $K_\omega$ is the aforementioned regenerative braking factor; $K_{accbasic}$ is the load matching factor, which can be expressed as follows:

$$K_{accbasic} = \begin{cases} 0 & \omega_k < 0 \\ \frac{\omega_k}{C_2\sqrt{T_{basic}}} & 0 < \omega_k < C_2\sqrt{T_{basic}} \\ 1 & C_2\sqrt{T_{basic}} < \omega_k \end{cases}, \tag{18}$$

where $C_2$ is a fixed coefficient.

On trimming Equation (17), we obtain the expression of the predicted speed:

$$\omega_{k+1} = \omega_k + T_s\left(\frac{T_m - K^2{}_{accbasic}(T_{basic} + K_\omega)}{(J_s - J_m)}\right). \tag{19}$$

Based on the inertia simulation predictive model, the predicted speed is parsed and input into the control loop of the load simulation algorithm as displayed in Figure 6. The PI controller parameters of the inner current loop and the outer speed loop are tuned by calculating the bandwidth of the current loop and the damping coefficient of the speed loop.

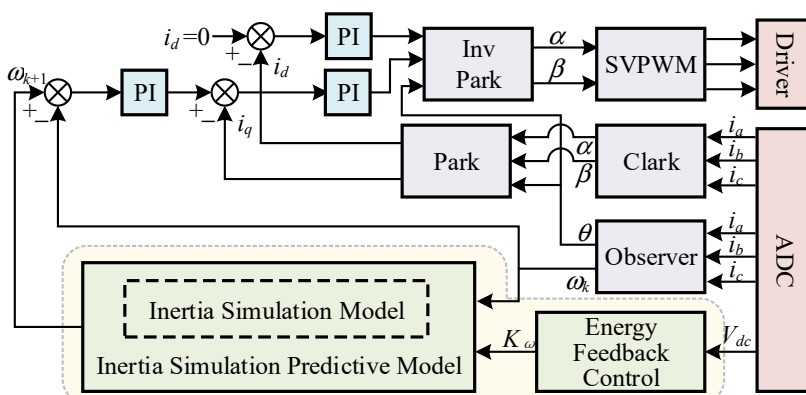

**Figure 6.** The complete block diagram of the load simulation algorithm.

Assuming that the current bandwidth is 20 times lower than the sampling frequency of the system, we have the following equation:

$$BW_i = \frac{2\pi F_s}{20}, \tag{20}$$

where $F_s$ is the sampling frequency of the system.

Combining Equations (20) and (21), the proportional coefficient $iK_p$ and the integral coefficient $iK_i$ in the current loop can be calculated as follows:

$$\begin{cases} iK_p = BW_i \cdot L_s \\ iK_i = \dfrac{R}{L_s} \end{cases}. \tag{21}$$

Because the speed controller is in the mechanical domain, which has considerably slower time constants where phase delays can be tighter, system stability is considerably affected. When tuning the speed loop, the moment of inertia is also a crucial parameter to be considered in the controller.

Combining Equation (22), the proportional coefficient $spdK_p$ and the integral coefficient $spdK_i$ in the speed loop are obtained as follows:

$$
\begin{cases}
spdK_p = \dfrac{4(J_s - J_m)}{3p_n\lambda_r\delta\tau} \\
spdK_i = \dfrac{1}{\delta^2\tau}
\end{cases}, \tag{22}
$$

where $\tau$ is the time constant of velocity filter; $\lambda_r$ is the back-emf coefficient; $\delta$ is the damping coefficient. If $\delta$ decreases, the bandwidth of the speed loop increases, which results in considerable overshoot. This phenomenon results in a longer time for speed rise and stability. When $\delta > 1$, the system phase margin is typically greater than 0, which renders the system stable. When $\delta$ is close to 1, severe underdamping performance typically results in oscillations.

## 6. Simulation and Experimental Verification

### 6.1. Simulation and Analysis

The inertia simulation model, the inertia simulation predictive model, and the load simulation algorithm based on the two models were simulated in MATLAB/Simulink to verify their effectiveness. A nonsalient-pole PMSM model was considered as the loading motor model. The parameters of this model are presented in Table 1. In the model of the bearing motor, an additional PMSM equipped with an independent speed regulation system is required to output the periodic driving torque in the simulation. The start and end moments of load simulation are $t = 1$ s and $t = 10$ s, respectively. When $t = 6$ s, the bearing motor removes the driving torque, and the loading motor runs continuously under the simulated inertial load. Other simulation parameters are as follows: the simulated load inertia is 5.06 kg·m$^2$; the preset speed is 145 r/min.

**Table 1.** The parameters of the motor.

| Parameter | Value |
| --- | --- |
| Number of pole pairs $p_n$ | 16 |
| Stator resistance $R$ ($\Omega$) | 0.38 |
| $d$-axis inductance $L_d$ (H) | 0.001315 |
| $q$-axis inductance $L_q$ (H) | 0.001315 |
| Permanent magnet flux linkage $\psi_f$ (Wb) | 0.4425 |

The simulation results are displayed in Figure 7. Figure 7a,b respectively detail the speed curve and load torque waveform of the loading motor before and after the application of this method; Figure 7c displays the speed fluctuation of the loading motor at a steady state from $t = 3$ s to $t = 5$ s, before and after the application of this method; Figure 7d details the torque ripple of the loading motor at a steady state from $t = 3$ s to $t = 5$ s, before and after the application of this method.

Figure 7 reveals that before using the aforementioned algorithm, the maximum speed fluctuation range of the loading motor is approximately 30, and the RMS of the speed fluctuation waveform is 6.953 at a steady state in the $t = 3$–5 s period. Furthermore, the maximum range of the torque ripple is close to 4, and the RMS of the torque ripple waveform is 0.781 at a steady state in the $t = 3$–5 s period. After utilising the aforementioned algorithm, the maximum speed fluctuation range of the loading motor is approximately 5, and the RMS of the speed fluctuation waveform is 1.106, which is 84% lower than before using the aforementioned algorithm when it is at a steady state in the $t = 3$–5 s period. The maximum range of the torque ripple is approximately 0.8, and the RMS of the torque ripple

waveform is 0.281, which is 64% lower than before using the aforementioned algorithm when it is at a steady state in the $t = 3$–$5$ s period.

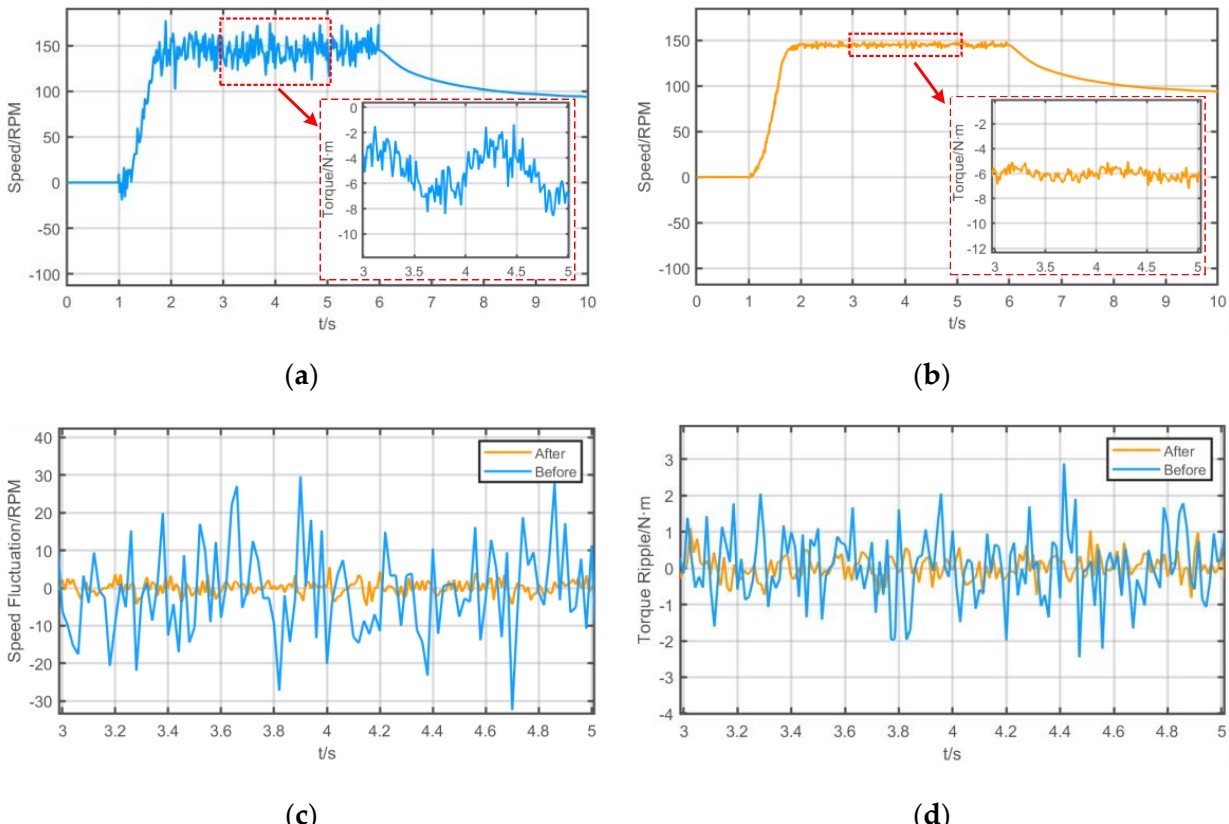

**Figure 7.** The simulation results. (**a**) The speed curve and load torque waveform of the loading motor before the application of this method. (**b**) The speed curve and load torque waveform of the loading motor after the application of this method. (**c**) The speed fluctuation of the loading motor at a steady state from $t = 3$ s to $t = 5$ s, before and after the application of this method. (**d**) The torque ripple of the loading motor at a steady state from $t = 3$ s to $t = 5$ s, before and after the application of this method.

Under the same simulation conditions, the simulation results revealed that the predictive model described in this study can calculate the predicted speed of the loading motor during the loading process based on the aforementioned inertia simulation model so that the loading process is not susceptible to strong-coupling torque interference, which is consistent with the dynamic characteristics of the target system under the same working conditions, and the inhibitory effect on the load torque ripple is considerable.

## 6.2. Experiments and Analysis

To verify the described method, a comparative experiment was performed on a self-developed experimental platform. TMS320F28069, a DSP of Texas Instruments, was used as the controller. A PMSM equipped with an independent speed regulation system was used as the bearing system. The bearing motor outputs periodic driving torque to drive the loading motor. The torque transfer can occur between the bearing motor and loading motor only through mechanical connection. The parameters of the loading motor are the same as in Table 1. The simulated load inertia is 5.06 kg·m$^2$. The experimental platform is displayed in Figure 8.

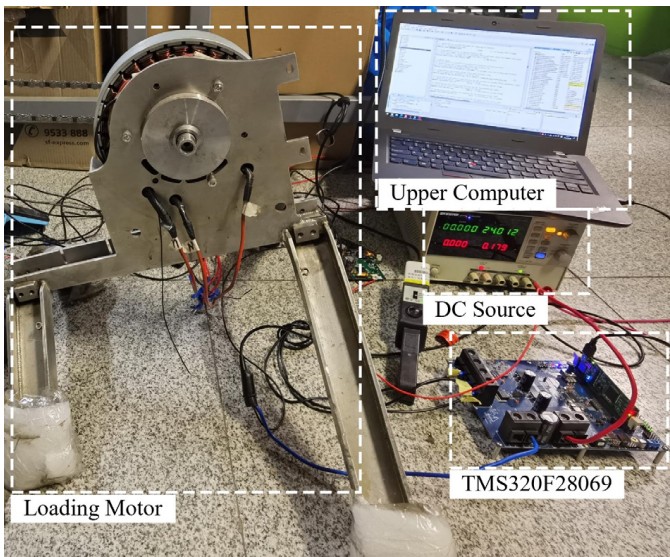

**Figure 8.** The experimental platform.

Figure 9a,b display the preset speed of the bearing motor and the actual speed of the loading motor before and after the application of this method, respectively. The aforementioned speed curves detail the degree of 'synchronous motion' between the loading motor and the bearing motor when simulating the load inertia.

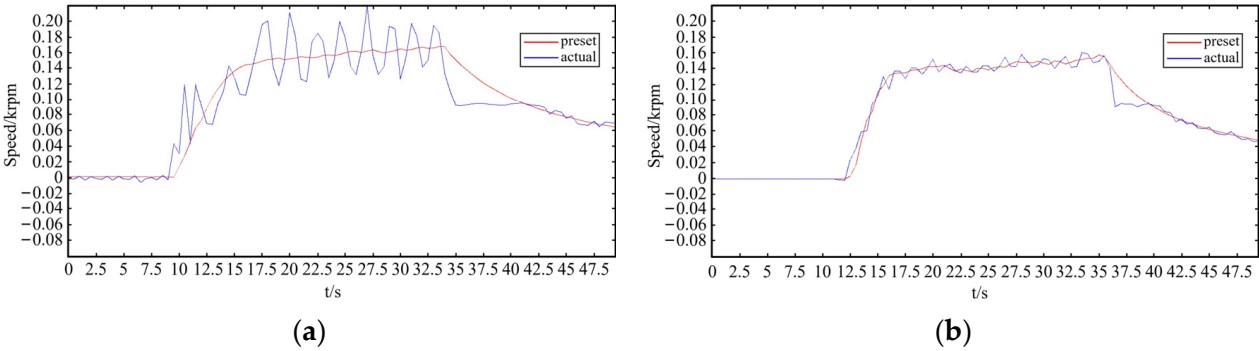

**(a)** **(b)**

**Figure 9.** (**a**) The preset speed of the bearing motor and the actual speed of the loading motor before the application of this method. (**b**) The preset speed of the bearing motor and the actual speed of the loading motor after the application of this method.

As displayed in Figure 9a, the actual speed of the loading motor fluctuates considerably around the preset speed of the bearing motor with the maximum speed fluctuation range of 46.8 at a steady state in the $t = 16$–34 s period; At $t = 34$ s, the bearing motor removes the drive torque, and the loading motor runs continuously under the simulated inertial load. At this point, the actual speed of the loading motor drops to 93 r/min with a decrease of 74 r/min, which then converges to the preset speed of the bearing motor after 7 s.

Figure 9b displays the result under the same working conditions. The maximum fluctuation range of the actual speed of the loading motor is approximately 7.5, which is 84% lower than before application of the aforementioned algorithm when it is at a steady state in the $t = 16$–35.5 s period. The bearing motor removes the drive torque at $t = 35.5$ s, which causes the actual speed of the loading motor to decrease to 92 r/min with a decrease of 63 r/min, which is 14.86% lower than before using the aforementioned algorithm. The speed then converges to the preset speed of the bearing motor after 4 s, which is 42.86% less than before.

Figure 10a,b display the load torque waveform of the loading motor at a steady state from $t = 20$ s to $t = 30$ s, before and after the application of this method, respectively. The

figures reveal that after using the aforementioned algorithm, the maximum range of the torque ripple is approximately 1.5, which is 75% lower than its value before the application of the aforementioned algorithm, which is approximately 6.

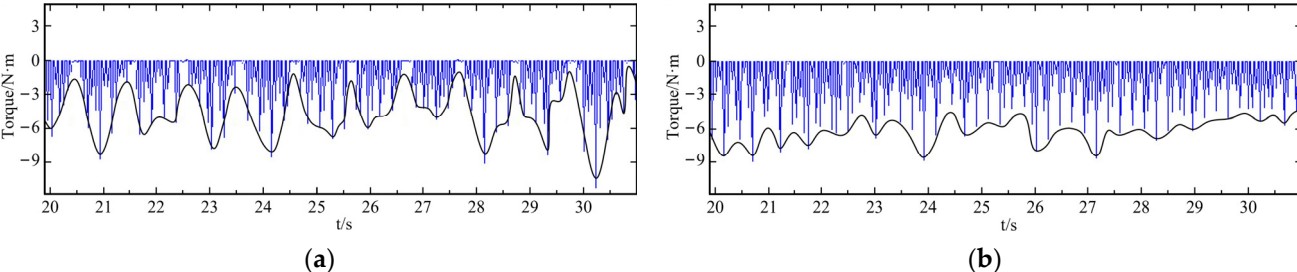

**Figure 10.** (**a**) The load torque waveform of the loading motor at a steady state from $t = 20$ s to $t = 30$ s, before the application of this method. (**b**) The load torque waveform of the loading motor at a steady state from $t = 20$ s to $t = 30$ s, after the application of this method.

Combined with the analysis in Figures 9a and 10a, the strong-coupling torque disturbance that is introduced by the loading motor responds to the periodic driving torque, which renders the 'synchronous motion' of the loading motor with the bearing system difficult, resulting in the fluctuation of the load torque output. Therefore, the smooth load torque cannot be output.

Combined with the results of the analysis in Figures 9b and 10b, the predicted speed based on the predictive model satisfies the dynamic characteristics of the target system under the same working conditions to realise the synchronous motion. The strong-coupling torque and load torque fluctuation caused by the position disturbance between the servo system and the loading system were considerably suppressed. The experimental results reveal that after using the aforementioned algorithm, the maximum speed fluctuation range of the loading motor decreases by 84% at a steady state, which is consistent with the simulation results. Furthermore, the maximum range of the torque ripple decreases by 75%, which conforms to the simulation results.

Figure 11a,b display the bus voltage of the system before and after this method is applied. Combining Equation (15) with the previous analysis revealed that because of the effect of the load inertia value, when $i_q < -\omega\psi_f/R$, $P_{in} > 0$, which indicates that the loading motor operates in the state of energy consumption braking and requires additional positive input power. When the motor operates in the state of energy consumption braking, an additional input power results in a decrease in the bus voltage, which affects EDLSS stability. In this experiment, when the bus voltage decreases to less than 36 V, the EDLSS is in a critical state and cannot maintain the normal operating voltage of the processor and the peripheral circuit.

As displayed in Figure 11a, during the $t_1$–$t_2$ period, both the loading motor and the bearing motor were in a standby state, and the bus voltage of the system was 24 V; during the $t_2$–$t_5$ period, when the loading motor was dragged by the bearing motor and simulated the aforementioned load inertia, the bus voltage should have increased and maintained its value at 83.2 V. However, because of the influence of the energy consumption braking state, the bus voltage decreased 12 times in 7 s. At $t_4$ and $t_5$ moments, the bus voltage decreased sharply to 36 V. At the $t_6$ moment, because of the braking state of energy consumption, the bus voltage could not maintain the normal operating voltage.

Utilising the proposed method to ensure the motor runs under the same working conditions, an oscilloscope was used to measure the bus voltage (Figure 11b). In the $t_3$–$t_5$ period, the jitter of the bus voltage decreased 7 times in 9 s. At the $t_4$ moment, through the effective control of the loading motor braking state, the bus voltage drop was limited to 52 V, which negatively affected the energy consumption braking state on EDLSS suppression.

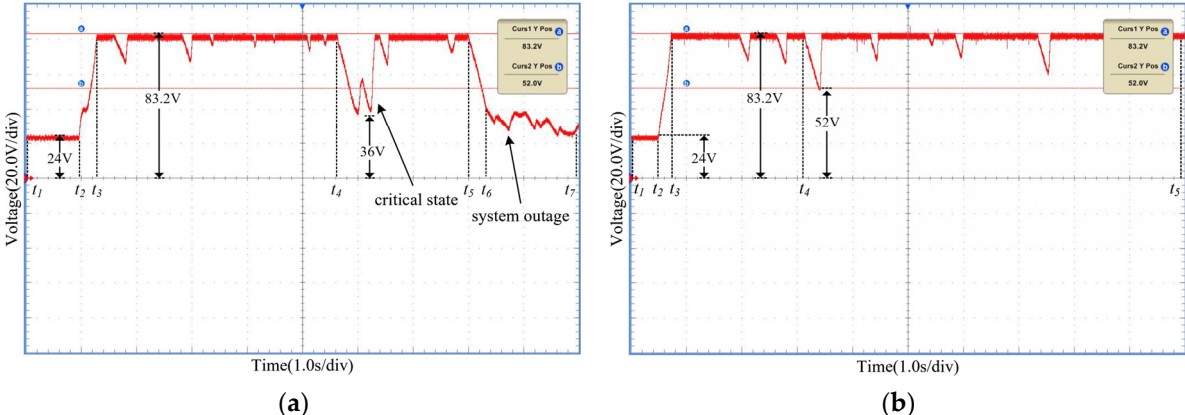

(**a**)　　　　　　　　　　　　　　　　　　　　　　　　(**b**)

**Figure 11.** (**a**) The bus voltage of the system before this method is applied. (**b**) The bus voltage of the system after this method is applied.

## 7. Conclusions

In this study, a load simulation algorithm based on an inertia simulation predictive model was proposed to mitigate strong-coupling torque disturbance and load torque fluctuation resulting from the motion state change of the bearing system in EDLSS. First, dynamic equations of both the EDLSS and the target system were combined to propose an inertia simulation model. Next, based on the inertia simulation model, an inertia simulation predictive model and a load simulation algorithm were proposed. Both the braking state and the power model of PMSM were analysed, and an energy feedback control method was proposed. The proposed method was simulated and experimentally verified. The following results were obtained:

(1) The inertia simulation model in this paper converted the conventional realisation method into the tracking of the motion characteristics of the target system under the same working conditions so that the loading process was not limited to the load torque and was not susceptible to the interference of strong-coupling torque. During the process of simulating the load and inertia torque, the proposed model paid more attention to the dynamic characteristics of the target system rather than the machine parameters, which made the dynamic characteristics of the simulation system consistent with the target system without a specific motor.

(2) The load simulation algorithm in this study analysed the predicted speed of the loading motor based on the predictive model so that the simulation system could satisfy the dynamic characteristics of the target system under the same working conditions. By realising synchronous motion, the strong-coupling torque and the load torque fluctuation caused by position disturbance of the servo system and the loading system were suppressed considerably.

(3) The energy feedback control method effectively controlled the braking state of the loading motor so that the adverse effects of its energy consumption braking state on the EDLSS were suppressed.

In conclusion, the proposed method can suppress the strong-coupling torque and load torque fluctuation considerably, as well as the adverse effects of the energy consumption braking state of the loading motor in the loading process of the EDLSS. With the exception of EDLSSs, this method has good reference value in servo system dynamic stiffness tests, mechanical back-to-back tests, and other applications.

**Author Contributions:** Conceptualization, W.W.; methodology, W.W., K.L., Z.W., X.L. and H.Z.; software, formal analysis, validation, K.L. and W.W.; writing—original draft preparation, K.L.; writing—review and editing, K.L. and W.W.; supervision, W.W.; project administration, W.W. All authors have read and agreed to the published version of the manuscript.

**Funding:** This research was funded by the National Natural Science Foundation of China, grant number 51977150.

**Institutional Review Board Statement:** Not applicable.

**Informed Consent Statement:** Not applicable.

**Data Availability Statement:** Not applicable.

**Conflicts of Interest:** The authors declare no conflict of interest.

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
