# Peer review of "A Dynamic Load Simulation Algorithm Based on an Inertia Simulation Predictive Model"

_applsci, doi:10.3390/app12147142_

Round 1

Reviewer 1 Report

1.      The reference style [1-3] and [13-18] is different from other citation style. Please make uniform and avoid using lumped references.

2.      The authors propose a load simulation algorithm and provide insightful results. Can authors give comparison between experimental and simulation results? How can authors claim their provided algo is better?

3.      Please improve the quality of Fig 11.

Author Response

Response to Reviewer 1 Comments

We are very grateful to the Reviewer for reviewing the article so carefully and raising these important points. We have carefully considered these suggestions and tried our best to improve our manuscript.

Here we respond to the Reviewer's comments:

Response 1: We fully agree with your suggestion that a unified citation style is more appropriate. We have revised the text to address your concerns and hope that it is now more rigorous (page 1, line 45, and page 2, lines 78, 95). Furthermore, we agree that lumped references is inappropriate. To better help researchers engaged in this field understand the current state of the research field, we classify the cited references according to the methods they use and summarize them first before introducing these methods. For example, [1318] are all from the perspective of the position error between the servo and the loading systems.

Response 2: We appreciate it very much for this great suggestion, and we have revised the manuscript to address your concerns (page 12, lines 389–393). Though at present, a variety of load simulation algorithms have emerged in current research, it is well accepted to calculate the required inertia torque according to the phase relationship between the position and acceleration signals measured by the sensor and track the given load torque. Under the same simulation and experimental conditions, the results reveal that the maximum speed fluctuation range of the loading motor, the RMS of the speed fluctuation, the maximum range of the torque ripple, and the RMS of the torque ripple are superior to the conventional algorithms. We sincerely hope that the answer can make you satisfied.

Response 3: We apologize for our error. It may seem that the PDF version of the manuscript compressed the figures' quality. We will provide a high-quality version later.

We would like to thank the Reviewer again for taking the time to review and improve our manuscript.

Reviewer 2 Report

Title: A Dynamic Load Simulation Algorithm Based on an Inertia 2 Simulation Predictive Model

Very nice work. Only minor comments:

The authors needs to explain for the reader whether or not this approach can be generalized for any type of machine.

It is important to highlight the main contribution of this article in the introduction, and what is new about it.

Author Response

Response to Reviewer 2 Comments

We are very grateful to the Reviewer for reviewing the article so carefully and raising these important points. We have carefully considered these suggestions and tried our best to improve our manuscript.

Here we respond to the Reviewer's comments:

Response 1: We fully agree with your suggestion and appreciate it very much. We have revised the manuscript to address your concerns and hope that it is now clearer (page 13, lines 432–436).

Response 2: We fully agree with your suggestion that the novelty and main contribution of the research should be highlighted. We have revised the text to address your concerns and hope that it is now clearer (page 2, lines 99–100, and page 3, lines 101–110).

We would like to thank the Reviewer again for taking the time to review and improve our manuscript.

Reviewer 3 Report

The article under review presents the dynamic load simulation system developed by the authors, proposes an inertia simulation predictive model and a load simulation algorithm, and suggests an energy feedback control method to improve the stability of the electric dynamic load simulation system.

The studies were carried out using the MATLAB/Simulink software, as well as on a specially designed laboratory experimental platform.

The results of the presented studies can be useful to scientists and researchers specializing in the field of electric drive systems.

During the review, I was left with a few questions and I drew attention to the following shortcomings, the correction of which would improve the quality of the article:

  1. The paper describes in detail the models of a non-salient pole permanent magnet synchronous motor, simulation and predictive models of inertia. This is widely known. I ask you to more clearly formulate the scientific achievements of the authors.
  2. Is there a practical application of the developments proposed by the authors? Please give a few examples. Do the authors of the paper have plans for further research?
  3. In my opinion, it does not look good when the paper has [4-12] (line 48), [8-12] (line 60), [13-18] (line 75).

Author Response

Response to Reviewer 3 Comments

We are very grateful to the Reviewer for reviewing the article so carefully and raising these important points. We have carefully considered these suggestions and tried our best to improve our manuscript.

Here we respond to the Reviewer's comments:

Response 1: We appreciate it very much for this great suggestion. The proposed novel inertia simulation model converts the conventional realisation method of the load simulation so that the loading process is not limited to the loading torque and is not susceptible to the interference of strong-coupling torque. The predicted speed calculated by the proposed novel predictive model is consistent with the dynamic characteristics of the target system under the same working condition so that the strong-coupling torque and the load torque fluctuation are suppressed considerably. We sincerely hope that the answer can make you satisfied.

Response 2: We take pleasure in introducing to you that the proposed algorithm can be used not only for electric dynamic load simulation systems but also for the aerodynamics load simulator, the electro-hydraulic load simulator, the mechanical back-to-back platforms, et cetera. Furthermore, when the bearing system has a complex transmission structure, nonlinear factors such as mechanical friction and transmission efficiency may harm the loading torque. Designing a novel extended state observer combined with this research is the next research direction. We sincerely hope that the answer can make you satisfied.

Response 3: We fully agree that lumped references is inappropriate. We have revised the text to address your concerns and hope that it is now more rigorous (page 2, line 51). To better help researchers engaged in this field understand the current state of the research field, we classify the cited references according to the methods they use and summarize them first before introducing these methods. For example, adaptive control methods were adopted in [812], and [1318] are all from the perspective of the position error between the servo and the loading systems.

We would like to thank the Reviewer again for taking the time to review and improve our manuscript.

Reviewer 4 Report

The article is devoted to developing an algorithm for an electrical dynamic load simulation system based on an inertia simulation prediction model to mitigate tight coupling torque disturbance and load torque fluctuations caused by a change in the motion state of a bearing system. The inertia simulation model was proposed by combining the dynamic equations of the electrical dynamic load simulation system and the target system. The inertial simulation model has transformed the traditional method of implementing inertial simulation into tracking the motion characteristics of the target system under the same operating conditions. The authors propose an inertial simulation prediction model and a load simulation algorithm based on the inertia simulation model, considering the effect of solid torque and motor braking state violation as two influential factors. The predicted speed calculated by the prediction model was consistent with the dynamic performance of the target system under the same operating conditions and input into the control loop. Based on the analysis of the braking state and the power model of a permanent magnet synchronous motor, the authors propose an energy feedback control method to improve the stability caused by the braking state of the load motor. The method proposed in the article was tested through simulation analysis and experimental justification on a self-developed experimental platform.

Despite the satisfactory quality of the article, some shortcomings need to be corrected.

  1. Expanding to abstract with actuality and numerical results of the study are recommended.
  2. The aim of the research should be defined.
  3. The known approaches should be separated from the ones proposed by the authors.
  4. The data used for the experimental investigation should be described in more detail.
  5. It is recommended to include the Discussion section to compare obtained results with other research.
  6. The scientific novelty of the research should be highlighted.
  7. Section 8 Patents is empty.

In summarizing my comments, I recommend that the manuscript is accepted after minor revision. 

Author Response

Response to Reviewer 4 Comments

We are very grateful to the Reviewer for reviewing the article so carefully and raising these important points. We have carefully considered these suggestions and tried our best to improve our manuscript.

Here we respond to the Reviewer's comments:

Response 1: We fully agree with your suggestion and appreciate it very much. We have revised the manuscript and added the actuality and numerical results of the study to Abstract (page 1, lines 29–33).

Response 2: We fully agree with your suggestion and appreciate it very much. We have revised the manuscript and highlighted the aim of the research in Section 1 Introduction (page 2, lines 99–100, and page 3, lines 101–110).

Response 3: We fully agree with your suggestion and appreciate it very much. In this study, we report a novel inertia simulation model and a novel predictive model, which are different from current research, and carried out the theoretical analysis and the methodology introduction, respectively. Under the same simulation and experimental conditions, the results reveal that the maximum speed fluctuation range of the loading motor, the RMS of the speed fluctuation, the maximum range of the torque ripple, and the RMS of the torque ripple are superior to the conventional algorithms. We sincerely hope that the answer can make you satisfied.

Response 4: We fully agree with your suggestion and appreciate it very much. We have revised the text to address your concerns and expanded more numerical results to Experiments and Analysis (page 12, lines 374–376).

Response 5: We appreciate it very much for this great suggestion. Though at present, a variety of load simulation algorithms have emerged in current research, it is well accepted to calculate the required inertia torque according to the phase relationship between the position and acceleration signals measured by the sensor and track the given load torque. We performed a comparative simulation and experiment under the same simulation and experimental conditions, then discussed the results in Section 6 Simulation and Experimental Verification. The results reveal that the maximum speed fluctuation range of the loading motor, the RMS of the speed fluctuation, the maximum range of the torque ripple, and the RMS of the torque ripple are superior to the conventional algorithms. We sincerely hope that the answer can make you satisfied.

Response 6: We fully agree with your suggestion that the novelty and main contribution of the research should be highlighted. We have revised the text to address your concerns and hope that it is now clearer (page 2, lines 99–100, and page 3, lines 101–110).

Response 7: We apologize for our error. We have now corrected this error and removed Section 8 Patents.

We would like to thank the Reviewer again for taking the time to review and improve our manuscript.
